# Occurrence and Development of Off-Odor Compounds in Farmed Hybrid Catfish (*Clarias macrocephalus* × *Clarias gariepinus*) Muscle during Refrigerated Storage: Chemical and Volatilomic Analysis

**DOI:** 10.3390/foods10081841

**Published:** 2021-08-09

**Authors:** Hatairad Phetsang, Worawan Panpipat, Atikorn Panya, Natthaporn Phonsatta, Manat Chaijan

**Affiliations:** 1School of Agricultural Technology and Food Industry, Food Technology and Innovation Research Center of Excellence, Walailak University, Nakhon Si Thammarat 80160, Thailand; hatairad.p@hotmail.com (H.P.); pworawan@wu.ac.th (W.P.); 2Food Biotechnology Research Team, Functional Ingredients and Food Innovation Research Group, National Center for Genetic Engineering and Biotechnology (BIOTEC), 113 Thailand Science Park, Phaholyothin Rd., Khlong Nueng, Khlong Luang, Pathumthani 12120, Thailand; atikorn.pan@biotec.or.th (A.P.); natthaporn.pho@biotec.or.th (N.P.)

**Keywords:** geosmin, 2-MIB, off-odor, hybrid catfish, lipid oxidation

## Abstract

The goal of this study was to examine the changes in chemical parameters, major volatile compounds, and sensory aspects in farm-raised hybrid catfish (i.e., dorsal, lateral line and ventral muscles) during a 15-day period of refrigerated storage. Trichloroacetic acid-soluble peptides, free fatty acid, total volatile base-nitrogen (TVB-N), and non-heme iron levels in all muscles increased as storage time proceeded. The levels of trans-1,10-dimethyl-trans-9-decalol (geosmin) and 2-methylisoborneol (2-MIB) were higher than their thresholds, which was connected to a stronger earthy odor. The concentrations of geosmin and 2-MIB in all muscles increased, although there was a consistent trend of earthy odor throughout storage; this phenomenon could be attributed to the masking effect of other off-odors. During storage, the largest lipid oxidation was found in ventral muscle, as measured by peroxide value and thiobarbituric acid reactive substances. During storage, the formation of the most volatile products increased in the lateral line and ventral muscle, whereas the dorsal muscle had the lowest concentration. As storage time proceeded, the strength of spoiled, fishy, rancid, and overall off-odor intensity of all tested muscles tended to rise. Those alterations were linked to higher levels of TVB-N and trimethylamine, as well as all other volatile lipid oxidation products (e.g., hexanal, propanal, 2,4 heptadienal, 1-octen-3-ol, octanal, nonanal, trans-2-heptenal, and 1-hexanol).

## 1. Introduction

One of Thailand’s most economically valuable farmed fish species is the hybrid catfish (*Clarias macrocephalus* × *Clarrias gariepinus*) [1]. In 2020, Thailand’s annual volume of farmed catfish is expected to be approximately 100,000 tonnes, with a market value of more than USD 150 million [1]. The availability of essential nutrients is connected to the benefits of consuming farmed catfish. Farmed catfish has been a rich source of high biological value protein [2,3] and polyunsaturated fatty acids (PUFA) [2,4]. Cahu et al. [5] found that farmed fish have higher levels of n-3 PUFA (particularly EPA and DHA) and α-tocopherol than wild fish. However, off-odor (e.g., earthy/fishy/rancid) is a major quality concern for farmed catfish fillet, and it is still a major problem in the catfish industry. One of the most prominent reasons for decreasing customer acceptance in fish and fish products is off-odor, which can render them unfit for sale [6]. Chemicals connected to off-odor are produced by microbial action, enzymatic activities, lipid oxidation, and environmentally/thermally driven reactions [7]. Trans-1,10-dimethyl-trans-9-decalol (geosmin) and 2-methylisoborneol (2-MIB) are the most common compounds that cause an earthy off-odor in fresh hybrid catfish mince [6]. Recent work revealed that not only geosmin and 2-MIB, but also (E)-2-nonenal and 1-octen-3-ol altered the earthy and muddy odor in farmed catfish muscles [6]. When considering earthy-musty odorants produced by the lipid oxidation reaction, Fu et al. [8] discovered that 1-octen-3-ol was one of the contributors to the earthy odor in the hemoglobin lipid oxidation system whilst adding hexanal or 1-octen-3-ol to an aqueous solution boosted the offensive intensity of 2-MIB and geosmin [9].

When fish specimens are cold stored for longer periods, more volatile lipid oxidation products are produced. The off-odor volatile chemicals (aldehydes, ketones, alcohols, acids, esters, and short-chain hydrocarbons) are typically formed during the lipid oxidation reaction, which imparts the fish flavor quality [10,11]. Hydroperoxide decomposition, which occurs when hydroperoxide is exposed to heme and non-heme iron or is exposed to high temperatures, produces such volatile lipid oxidation products [11,12]. Membrane breakdown allows unsaturated lipids to interact with enzymes and catalysts, including heme, non-heme iron, and other metal cations, potentially leading to the onset of lipid oxidation [13]. The distribution of lipid levels/lipid compositions, pro- and antioxidants, as well as availability to oxygen, differs between different muscle regions of fish [10]. The stronger pro-oxidants, such as iron, copper, and total aqueous prooxidative activity, are mostly responsible for the increased lipid oxidation in dark muscle of herring [14]. Silver carp belly flap muscle, on the other hand, has been primarily driven by increased total fat and PUFA instability to be the most vulnerable to lipid oxidation, with hexanol and 1-octen-3-ol as lipid oxidation indicators in its muscle [15]. Thus, the distribution of volatile lipid oxidation products in muscles would be different. Despite the fact that geosmin and 2-MIB are lipophilic compounds, their distribution is unaffected by fat content.

Cold storage of fish during transportation and processing is a frequent post-harvest management procedure to maintain the freshness of fish. Several indications have recently been used to assess the freshness and shelf-life of fish while being stored [16]. Recent advancements in fish freshness evaluation include colorimetric imaging [17], hyperspectral imaging [18], and sensing technologies [19]. Chemical measurements (such as total volatile base-nitrogen (TVB-N), trimethylamine (TMA), peroxide value (PV), thiobarbituric acid reactive substances (TBARS), free fatty acid (FFA), and K values) and sensory analyses, on the other hand, are still considered to be reliable indicators for estimating the shelf life of fish [16,20]. Chemical alteration, lipid oxidation, and the development of off-flavor volatile compounds nevertheless occurs in fish muscle during cold storage [21,22,23]. Thus, the chemical changes, volatile development, and unpleasant off-odor creation in cold-stored farmed hybrid catfish fillets, notably earthy and other off-odor types, were studied. The alterations in chemical characteristics, geosmin, and 2-MIB, as well as the major volatile lipid oxidation products, in the dorsal, lateral line, and ventral muscles of farmed hybrid catfish following a 15-day period of refrigerated storage were studied for the first time in this work. The aim of this study was to determine the chemical indices and distribution of odor-causing compounds in hybrid catfish muscles during refrigerated storage. The chemical and volatilomic information gathered can be used to develop a processing approach that allows all muscle types of farmed hybrid catfish to be consumed sustainably.

## 2. Materials and Methods

### 2.1. Fish Samples

Live farmed hybrid catfish (*C. macrocephalus* × *C. gariepinus*) with an average weight of 500–600 g were purchased from Thasala market, Nakhon Si Thammarat, Thailand. At the market, fish were stunned by an accurate blow to the head, which complies with animal welfare laws, and immediately packed in polystyrene foam boxes filled with ice (fish:ice ratio of 1:2, *w*/*w*) and brought to the laboratory within 20 min. Upon arrival, the fish were washed, headed, eviscerated, and filleted before being used. Individually packed in plastic bags, the fillets were kept at 4 °C for 15 days. Twenty fillets were chosen at random and excised into the dorsal, lateral line, and ventral muscles on days 0, 3, 6, 9, 12, and 15. To form a composite sample for the analysis, each muscle was minced in an MK 5087 M Panasonic Food Processor (Selangor Darul Ehsan, Malaysia). According to AOAC [24], the initial fat content of dorsal, lateral line, and ventral muscles was 5.3, 4.4, and 7.6 g/100 g, respectively. Dorsal, lateral line, and ventral muscles had initial total PUFA content of 22.4, 20.0, and 23.5 g/100 g total fatty acid, respectively.

### 2.2. Determination of Moisture, pH, and Trichloroacetic Acid (TCA)-Soluble Peptide Contents

Dorsal, lateral line and ventral muscle were analyzed for moisture using the method of AOAC [24]. After homogenizing fish muscle with 10 vol of deionized water (*w*/*v*) using an IKA Labortechnik homogenizer (Selangor, Malaysia), the pH was determined using a pH meter (Cyberscan 500, Singapore) [25].

For the TCA-soluble peptide, the sample was homogenized with 3 vol of 5% (*w*/*v*) TCA (Sigma–Aldrich Co., St. Louis, MO, USA) before being placed on ice for 1 h. The soluble peptides in the supernatant were measured and recorded as μmole tyrosine/g sample after centrifugation (RC-5B plus, Sorvall, Norwalk, CT, USA) at 5000× *g* for 5 min [26].

### 2.3. Measurement of FFA Content

The Bligh and Dyer method [27] was used to extract lipid prior to analysis. The extraction solvents used were chloroform and methanol. The method of Lowry and Tinsley [28] was used to estimate the FFA. A 0.1 g sample was dissolved in 5 mL of isooctane after vigorous swirling. The mixture was added to 1 mL of a 5% (*w*/*v*) cupric acetate-pyridine reagent, which was made by dissolving cupric acetate (Sigma–Aldrich, St. Louis, MO, USA) (5 g) in distilled water (100 mL), filtering, and correcting the pH to 6.0–6.2 with pyridine (Sigma Aldrich). The mixture was vigorously shaken for 90 s with a vortex mixer before being allowed to stand for 10–20 s. At 715 nm, the absorbance of the top layer was measured. Palmitic acid (Sigma–Aldrich) was used to create a standard curve with concentrations ranging from 0 to 10 μmol/mL. The amount of FFA in each lipid sample was calculated as g FFA/100 g lipid.

### 2.4. Determination of TVB-N and TMA Contents

The TVB-N and TMA content were determined following the Conway micro-diffusion method as described by Panpipat and Chaijan [26]. Briefly, 2 g of fish mince was homogenized with 8 mL of 4% TCA. The filtrate from the mixture that would be employed for analysis was filtered using Whatman No 41 filter paper. For TMA analysis, formaldehyde was added to the filtrate to fix the ammonia present in the sample. TVB and TMA were released after the addition of saturated K_2_CO_3_ and diffused into the boric acid solution. The solution was titrated, and the quantity of TVB-N or TMA was determined.

### 2.5. Measurement of Heme and Non-Heme Iron Contents

The heme iron content was determined using the method described by Benjakul and Bauer [29] and was reported in mg/100 g sample. The non-heme iron content was measured using the method of Schricker et al. [30] and was reported as mg/100 g sample.

### 2.6. Measurement of PV and TBARS

The PV and TBARS were determined by the method of Panpipat et al. [31]. The PV was measured in milliequivalents (meq) of free iodine per kg of sample. For TBARS analysis, 5 g fish mince was homogenized with 25 mL of TBARS solution (0.375% TBA, 15% TCA, and 0.25 N HCl) followed by 10 min of heating in boiling water to develop a pink color. The mixture was then cooled under running water and centrifuged at 5500× *g* for 25 min. The absorbance of the supernatant was measured at 532 nm. The TBARS was measured in mg of malondialdehyde (MDA) equivalent per kg of sample.

### 2.7. Determination of Geosmin, 2-MIB, and Volatile Lipid Oxidation Products

Geosmin, 2-MIB, and volatile lipid oxidation products were evaluated by Phetsang et al. [6] using the headspace-solid phase microextraction procedure in combination with GC-quadrupole-time-of-flight mass spectrometry. For extraction, a CWR/PDMS (120 μm) fiber was employed. The minced sample (3 g) was placed in a HS vial (20 mL), accompanied by NaC1 (1.5 g) and 2 mL of distilled water to make a final volume of 5 mL, and then tightly capped followed by incubation at 70 °C/50 min. The CWR/PDMS fiber was then placed in the GC injector port. For the gas chromatography–mass spectrometry study, a gas chromatograph (Agilent 7890B, Santa Clara, CA, USA), a mass spectrometer (Agilent 7250, Santa Clara, CA, USA), and an auto sampler system (PAL3, RTC, CTC Analytics AG, Switzerland) were used. A DB-WAX capillary column was used to evaluate the samples. High purity helium was used as the carrier gas. Temperatures in the oven were preset (45 °C/2.5 min, then raised at 10 °C/min to 80 °C, then increased at 10 °C/min to 250 °C, and maintained for 2 min). The MS spectra were recorded in electron ionization mode with a 70 eV ionization energy. The scan rate was 4.4 scan/s and the mass range was 35–350 *m*/*z*. The MS data was acquired using the scan mode, and the quantitation analysis was performed using the extraction of ion chromatograms. All of the identification parameters are shown in Table 1, including, volatile species, retention time, retention index, and selected ions (*m*/*z*) used for the qualitative and quantitative analysis. To produce calibration curves for the calculation of volatile compounds (geosmin, 2-MIB, propanal, hexanal, cis-4-heptenal, octanal, trans-2-heptenal, 1-hexanol, nonanal, 1-octen-3-ol, and 2,4-heptadienal), a standard addition method was used.

### 2.8. Off-Odor Evaluation

Six trained panelists sniffed the samples smoothed out on the bottom of Erlenmeyer flasks to assess the off-odor intensities of the dorsal, lateral line, and ventral muscles at day 0, 3, 6, 9, 12, and 15 of storage. The panelists were trained three times a week for a total of two weeks prior to the evaluation. Standard volatiles including geosmin combined with 2-MIB for earthy odor, TMA for fishy odor, oxidized fish oil for rancid odor, and farmed hybrid catfish stored in ice for 18 days for spoiled odor were used to train panelists. On a line scale ranging from 0 (none) to 4 (high), the intensity of fishy, earthy, rancid, spoiled, and overall off-odor was graded. Walailak University’s Human Research Ethics Committee authorized the experimental protocol (WUEC-21-045-01) and the sensory analysis was performed following the NIH guidelines [32].

### 2.9. Statistical Analysis

Significant variations (*p* < 0.05) across samples were analyzed using Duncan’s multiple-range test. Data analysis was conducted using SPSS 23.0 for Windows (SPSS Inc., Chicago, IL, USA).

## 3. Results and Discussion

### 3.1. Changes in Moisture Content

Lateral line muscle had the highest moisture content, followed by dorsal, and ventral muscle during 15 days of refrigerated storage (*p* < 0.05) (Figure 1a). Chaijan et al. [33] found that the moisture content of dark muscle was higher than that of ordinary muscle in catfish (*Clarias macrocephalus*). Similarity, Testi et al. [34] reported that dorsal muscle constituted more moisture than ventral muscle in sea bream, sea bass, and rainbow trout. The moisture content of lateral line muscle remained constant as storage time progressed (*p* > 0.05). For the dorsal and ventral muscles, the moisture content gradually decreased up to day 9 of storage, and then gradually increased afterwards (*p* < 0.05). However, the dorsal and ventral muscles’ final moisture content was lower than the starting levels (*p* < 0.05). These results indicated that dorsal and ventral muscles had a lower water holding capacity (WHC) when compared to lateral line muscle. Sun et al. [35] reported that, as storage time passed, the binding of water in fish loosened, and some of the free water was pressed to the fish’s surface, suggesting a decrease in WHC.

### 3.2. Changes in pH

The initial pH values of dorsal, lateral line, and ventral muscle were 6.58, 6.55. and 6.54, respectively (Figure 1b). All three muscles showed the similar fluctuating manner in pH values during refrigerated storage for 15 days. The pH of all three muscles dramatically decreased up to day 6 of storage, gradually increased up to day 9, decreased again at day 12 and sharply increased afterwards (*p* < 0.05). The decrease of pH during the first period of storage may be influenced by lactic acid from glycolysis, proton from ATP hydrolysis, and microbial action [36]. After day 6, the pH began to fluctuate, with an increasing trend toward the end. This was most likely due to the possibility of both acid and alkaline compounds forming during storage. The pH can then be adjusted to match the rate of acid and alkaline production. The increase in pH at the end of storage was associated with an increase in volatile bases produced by endogenous or microbial enzymes [37]. This was in agreement with Rawdkuen et al. [38] who found that the pH of dorsal and ventral muscles from giant catfish (*Pangasianodon gigas*) increased as the storage period progressed.

### 3.3. Changes in TCA-Soluble Peptide and FFA Contents

TCA-soluble peptides content in dorsal, lateral line, and ventral muscles were 10.78, 9.31, and 18.33 μmol tyrosine/g on the first day of storage, respectively (Figure 1c), indicating the existence of endogenous oligopeptides as well as breakdown products formed during post-harvest management [39]. TCA-soluble peptides in refrigerated storage of all studied muscles tended to increase when storage time progressed, suggesting that autolysis and microbial proteolysis continuously occurred during storage. In comparison to the other two muscles, the ventral muscle had the highest TCA-soluble peptide concentration during storage, indicating stronger protease activity. Endogenous or microbial proteases were found to be a possible source of proteolytic muscle breakdown during cold storage [40]. However, contamination of visceral proteinases during fish evisceration was one of the major causes to the higher proteolysis in the belly muscle [41]. Furthermore, this outcome was closely associated with the change in moisture content in the ventral muscle (Figure 1a). The lowest water-protein interaction in the ventral muscle corresponded to the highest protein degradation during refrigerated storage. Some low-abundance proteins, such as α-actinin, desmin, and dystrophin, degraded during storage, exposing hydrophobic residues and altering protein structure, which can modify the binding ability of protein and water [35]. At the end, all muscles had the highest TCA-soluble peptide concentration, signifying muscle breakdown and protein degradation.

The amount of FFA in all muscles increased with storage time (*p* < 0.05) (Figure 1d), demonstrating that lipid hydrolysis in farmed hybrid catfish muscles progressed during refrigerated storage. The hydrolysis of glycerol-fatty acid esters in postmortem fish muscle lipids is one important alteration that happens with the release of FFA [38]. Lipases and phospholipases are enzymes that effectively break down fish lipids [42]. Throughout storage, the lateral line had the highest FFA concentration, followed by ventral and dorsal muscles, respectively. Lipases are found in the digestive tract of fish (such as the liver, gut, stomach, and pyloric caeca) and muscle tissues [43]. Red muscle (e.g., lateral line) had the highest lipase activity among the muscle types studied [22]. Finally, a significant rise in FFA content of the lateral line muscle was observed (*p* < 0.05).

Throughout storage, the dorsal muscle had the lowest FFA content (*p* < 0.05). The low fat content, as well as lipase and phospholipase activity in dorsal muscle, may have contributed to this result. Despite the fact that the ventral muscle has the highest fat content, it has a lower FFA concentration than the lateral line muscle. This was related to the increased lipase activity in the lateral line, as was previously mentioned. It could also be owing to FFA’s susceptibility to lipid oxidation in this muscle.

### 3.4. Changes in TVB-N and TMA Contents

During refrigerated storage, the values for TVB-N increased as time progressed (*p* < 0.05). (Figure 1e). The TVB-N content of all muscles increased substantially after storage on days 0–3, minimally increased on days 3–9, and then climbed fast again on day 12 (*p* < 0.05). After that, the TVB-N of the dorsal muscle and lateral line tended to stabilize, whereas the TVB-N of the ventral muscle continued to rise (*p* < 0.05). During storage, TVB-N can be an indirect indicator of microbiological deterioration. A TVB-N value of 30 to 40 mg/100 g has been reported as the acceptable limit for temperate and cold water fish [44]. According to the findings, the TVB-N value of ventral muscle (31.50 mg/100 g) was unacceptable at day 15. The development of TVB-N and TMA is linked to microbial growth and can be utilized as a spoiling indicator. These two characteristics have also been linked to a fishy odor [45]. The TMA increased as storage time progressed (*p* < 0.05) (Figure 1f). TMA as well as TVB-N were highest in the ventral muscle at the last stage of storage. This could be caused by visceral contamination in the ventral muscle because enzymes and bacteria can be generally found in the viscera. Trimethylamine oxide (TMAO) is commonly used as an electron acceptor in anaerobic respiration by spoilage bacteria, such as *Shewanella putrefaciens*, *Photobacterium phosphoreum*, and Vibrionaceae, resulting in off-odor and off-flavor due to TMA formation [46]. When compared to marine fish, the presence of TMAO in freshwater fish is less common, but it does exist [47]. According to this study, TMA levels in all of the tested muscles were below the acceptable range of 5 mg/100 g after 15 days of refrigerated storage [48].

### 3.5. Changes in Heme Iron and Non-Heme Iron Contents

Throughout storage, the lateral line exhibited a larger heme iron level than other muscles (*p* < 0.05) (Figure 2a). In normal post-mortem muscle, such changes could be attributable to the blood residual differential between white and red muscles [30,49]. Iron is mainly derived from myoglobin and hemoglobin [50], indicating that lateral line muscle has more of myoglobin and hemoglobin. The heme iron content of farmed hybrid catfish muscles decreased during refrigerated storage (Figure 2a), likely due to the liberation of free iron from heme proteins to generate non-heme iron (Figure 2b) [49]. The heme iron content of all muscles stayed stable for the first 6 days of storage (*p* > 0.05) and then began to drop (*p* < 0.05). When compared to the original value at day 0, the heme iron content of every muscle reduced by 30–40% at the end. The dorsal muscle has the most heme protein degradation (40%) followed by the lateral line (34%) and ventral muscles (29%) respectively. The largest non-heme iron content at the end of storage was connected with the higher heme iron degradation rate in the dorsal muscle (Figure 2b). With increased storage time, non-heme contents from the dorsal, lateral line, and ventral muscles tended to rise (Figure 2b), which corresponded to a decrease in heme iron content (Figure 2a). In general, an increase in non-heme iron content is affected by a considerable increase in free iron release from a muscle that has been extensively degraded [49]. The results indicated that the porphyrin ring was likely disrupted during storage, resulting in the release of non-heme iron. After 6 days of storage, the non-heme iron content of the lateral line and ventral muscles increased dramatically, progressively declined until day 12, and then surged sharply again (*p* < 0.05). While the dorsal muscle showed a minor decline after 3 days of storage, a significant increase at day 6, a drastic decrease at day 9, and finally a rapid increase. The rate of accumulation and subsequent participation in deteriorative reactions caused changes in non-heme iron content during storage. Because non-heme irons are prooxidants, they can cause and catalyze a variety of deteriorative reactions, especially lipid peroxidation and the Maillard browning reaction [49,51,52].
Figure 1Changes in moisture content (**a**), pH (**b**), TCA-soluble peptide (**c**), free fatty acid (**d**), total volatile base-nitrogen (TVB-N) (**e**), and trimethylamine (TMA) (**f**) of farmed hybrid catfish dorsal muscle (―●―), lateral line (---■---), and ventral muscle (⋯▲⋯) during 15 days of refrigerated storage. The bars indicate standard deviation from triplicate determinations.
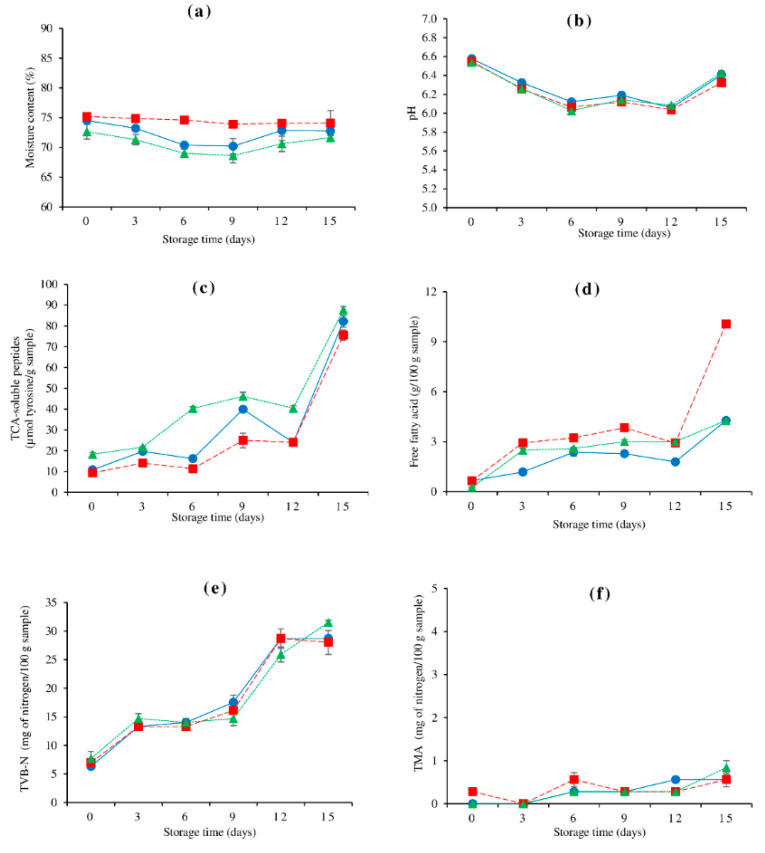



### 3.6. Changes in PV and TBARS

The PV value was used to assess the generation of hydroperoxide during the early stages of lipid oxidation (Figure 2c). All muscles had identical initial PV (*p* > 0.05). After 3 days of storage, the PV of the ventral muscle surged significantly, then sharply reduced until day 9, when it steadily increased. PV increased considerably in the dorsal muscle for the first 6 days, then quickly reduced on day 9 and remained constant thereafter (*p* > 0.05). PV of the lateral line muscle, on the other hand, reduced marginally after 3 days of storage (*p* < 0.05) and thereafter remained constant (*p* > 0.05). The variations in PV were caused by the instability of peroxides, which can be converted to other secondary products (e.g., aldehydes, ketones, and alcohols) [53]. As a result, the level of detection was determined by the degree of formulation and decomposition. The substantial increase in TBARS until the end of storage was strongly associated with the quick PV increase in the ventral muscle after 3 days of storage, therein showing the gradation of hydroperoxide to secondary lipid oxidation products. Finally, the ventral muscle showed the highest PV, followed by the lateral line and the dorsal muscle.

The concentration of polar secondary reaction products, particularly aldehydes, has been determined using TBARS [22]. The initial values of TBARS in the dorsal, lateral line, and ventral muscles were 0.69, 0.82, and 0.57 mg MDA/kg, respectively (Figure 2d), indicating that some lipid oxidation occurred during post-harvest handling. Some endogenous antioxidants, such as tocopherols, carotenoids, peptides, and other non-protein nitrogenous substances, may have aided in delaying lipid oxidation in all muscles during the storage period [33,49]. TBARS in the ventral muscle increased for up to 12 days before rapidly decreasing (*p* < 0.05). Furthermore, at days 3–12, the ventral muscle had a larger TBARS level than the other two muscles. The ventral muscle had the highest degree of lipid oxidation as measured by TBARS, which could be linked to the most muscle protein breakdown as measured by TCA-soluble peptides (Figure 1c). In the disintegrating muscle, oxygen penetration and the potential to interact with muscle lipids should be increased. Although the concentration of pro-oxidant, non-heme iron, in all muscles was similar during storage (Figure 2b), the possibility of pro-oxidant and unsaturated fatty acid interaction may be higher in the ventral muscle, which was more deteriorated. Furthermore, the ventral muscle exhibited the largest lipid content (7.6 g/100 g) and total PUFA content (23.5 g/100 g total fatty acid), indicating that it can trigger greater oxidation. The silver carp belly flap has the greatest TBARS, according to Kunyaboon et al. [15], due to its high lipid content and PUFA instability. Moreover, the presence of visceral lipoxygenase, peroxidase, and microsomal enzymes could play a crucial role in the oxidation of lipid [54]. The lipoxygenase activity of fresh silver carp ventral muscle was also shown to be higher than that of dorsal muscle [55]. The reactivity of MDA with amino acids, sugar, and other chemicals in complexes resulted in a significant decrease in TBARS of the ventral muscle at the end of storage [56]. Myofibrillar proteins can cross-link with MDA, causing structural and functional alterations [57]. The highest level of TBARS in the dorsal muscle was discovered on day 6 of storage and thereafter steadily dropped (*p* < 0.05). Li et al. [55] observed that the dorsal muscle of salted silver carp had a lower TBARS than the ventral muscle. According to Li et al. [55], the larger the protein content of the dorsal muscle, the more likely it is to crosslink with MDA.

In comparison to the other two muscles, the TBARS tended to stay at the same level on the lateral line muscle. The lateral line muscle with the highest moisture content (Figure 1a) has a lower protein degradation rate (Figure 1c), which may help to maintain the muscle’s lipid oxidative stability. Throughout storage, the lateral line muscle has kept the most moisture content (Figure 1a) and the least amount of TCA-soluble peptides (Figure 1c). Intact muscle may experience oxygen supply restrictions. In muscles with greater moisture content, the pro-oxidant may be diluted. Despite having the highest FFA (Figure 1d) and heme iron concentration (Figure 2a), the PV and TBARS of the lateral line muscle did not alter significantly after storage (*p* > 0.05). This could be related to the interaction between FFA and heme proteins, which results in the production of hemichrome; since it was discovered that hemichrome reduced heme proteins’ ability to oxidize lipids [58].
Figure 2Changes in heme iron (**a**), non-heme iron (**b**), peroxide value (PV) (**c**), and thiobarbituric acid reactive substances (TBARS) (**d**) of farmed hybrid catfish dorsal muscle (―●―), lateral line (---■---), and ventral muscle (⋯▲⋯) during 15 days of refrigerated storage. The bars indicate standard deviation from triplicate determinations.
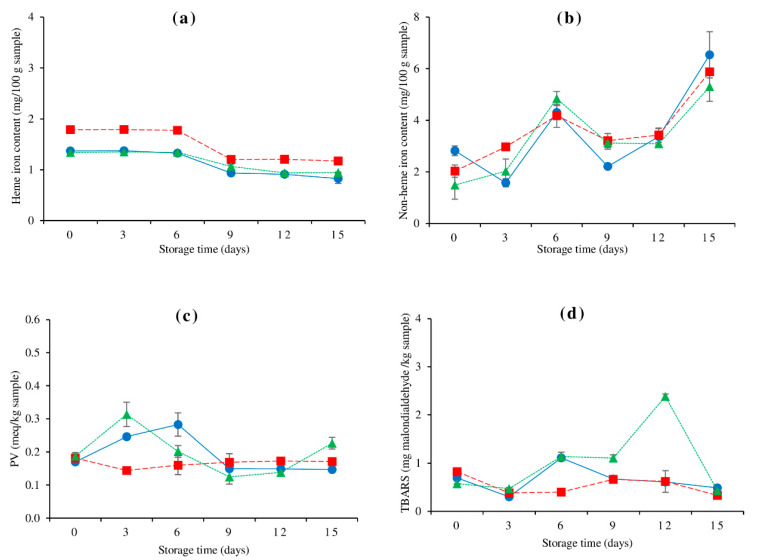


### 3.7. Changes in Geosmin and 2-MIB Contents

The cyanobacteria metabolites geosmin and 2-MIB accumulate in lipid-rich fish tissue [59]. Figure 3 shows the variations in geosmin and 2-MIB in the dorsal, lateral line, and ventral muscles of farmed hybrid catfish after 15 days of refrigerated storage. The initial amounts of geosmin (Figure 3a) and 2-MIB (Figure 3b) in all muscles was found to be higher than their thresholds (0.1–0.2 ng/g and 0.25–0.5 ng/g, respectively) [60], which correlated with the increased intensity of earthy odor compared to other off-odor types (see Section 3.9). The concentrations of geosmin and 2-MIB in all of the muscles tested increased with storage time (*p* < 0.05). Furthermore, during storage, the 2-MIB of the dorsal muscle increased considerably. Because geosmin is integrally connected with membrane lipids, but 2-MIB favors neutral lipids, DeWitt et al. [61] hypothesized that 2-MIB removal in protein isolates was easier than geosmin removal. Protein degradation may enhance the release of geosmin and 2-MIB from the fish matrix into the headspace, meaning that geosmin and 2-MIB have a stronger influence on stored hybrid catfish muscle off-odor. Since then, changes in TCA-soluble peptides have revealed that protein breakdown in farmed hybrid catfish muscles is proportional to storage time (Figure 1c). When compared to the fish protein isolate, Yarnpakdee et al. [62] found that the fish protein hydrolysate had greater geosmin release and recovery.

### 3.8. Changes in Lipid Oxidation-Derived Volatile Compounds

Figure 4 shows the effects of storage time and muscle location on selected volatile compounds in farmed catfish muscle. Except for cis-4-heptenal, the generation of all volatile compounds increased in the lateral line and ventral muscles after 12 or 15 days of storage (*p* < 0.05). Subsequently, most of those volatile chemicals declined considerably at the end of storage, which corresponded to a drop in TBARS values (Figure 2d). The interactions between the aldehyde and the amine and sulfhydryl groups lowered the hexanal volatility of washed turkey muscle proteins, according to Pignoli et al. [63]. Aldehyde lipid oxidation products can bind to myofibrillar proteins in fish muscle, according to Chaijan et al. [64]. Meanwhile, except for octanal, trans-2-heptenal, and cis-4-heptenal, the dorsal muscle had a constant trend in the generation of all volatile lipid oxidation products with the lowest intensity throughout storage. This finding showed that during 15 days of refrigerated storage, the dorsal muscle of hybrid catfish had the lowest concentration of undesired compounds from lipid oxidation.

Throughout storage, the largest quantities of propanal, hexanal, and 2,4 heptadienal were observed in the ventral muscle. According to our earlier study, n-3 linolenic acid was only found in the ventral muscle of hybrid catfish. Propanal and 2,4 heptadienal are oxidation products of n-3 linolenic acid [65]. Thus, the presence of their parent compounds was strongly connected with the considerably higher levels propanal and 2,4 heptadienal in the ventral muscle. Meanwhile, hexanal is produced when n-6 linoleic acid is oxidized [66]. According to our earlier study, the ventral muscle of farmed hybrid catfish had the highest levels of n-6 linoleic acid. The highest quantities of propanal and hexanal in the ventral muscle corresponded to the muscle’s highest TBARS on day 12 of storage. Propanal and hexanal were the appropriate lipid oxidation markers of farmed hybrid catfish muscle during refrigerated storage for 15 days, according to this finding. Due to their stable structure induced by the lack of a double bond, hexanal and propanal are commonly utilized as lipid oxidation indicators in headspace techniques [67]. Chaijan et al. [68] found that the off-flavor generation of chilled Asian sea bass steaks was substantially connected with propanal concentration and TBARS value. One of the most dependable indications of fish off-flavor is propanal [69]. Hexanal, on the other hand, was found to be an index of flavor deterioration in stored meat by Shahidi and Pegg [70]. When compared to the ventral muscle, the lesser generation of propanal, hexanal, and 2,4 heptadienal in the lateral line muscle was most likely owing to a lower concentration of fat and their parent substances. At the end of storage, the lateral line muscle had the highest levels of octanal, 1-octen-3-ol, and nonanal (day 12–15) (*p* < 0.05). Furthermore, 1-hexanol production was higher in the lateral line muscle than in other muscles, and it grew rapidly throughout the storage period (*p* < 0.05). These findings suggested that an abundant heme protein system affected the synthesis of octanal, 1-octen-3-ol, nonanal, and 1-hexanol. The hemoglobin-induced lipid oxidation mechanism in silver carp mince produced 1-octen-3-ol and nonanal, according to Fu et al. [8]. In Asian seabass muscle, autoxidation by blood produced 1-hexanol, nonanal, and octanal [71]. When compared to unbled Asian seabass, a lower quantity of heme and non-heme iron may reduce the generation of heptanal, hexanal, octanal, nonanal, and nonenal, as well as the fishy odor [71]. Except for octanal and cis-4-heptenal, the lowest concentration of volatile lipid oxidation products was seen in the dorsal muscle throughout storage. As a result, the dorsal muscle might be used as a substitute for surimi or other protein-based manufacturing that requires the raw material to be kept chilled for an extended period of time. In addition, all of the muscles studied showed a consistent trend in cis-4-heptenal content. Throughout the storage, the dorsal muscle had a 3-fold higher concentration of cis-4-heptenal than the other two muscles. Since cis-4-heptenal is produced from n-3 PUFAs, it has been identified as one of the major volatiles involved in rancid odor [72].

### 3.9. Changes in Sensory Scores

Figure 5 depicts changes in off-odor intensities in the dorsal, lateral line, and ventral muscles of farmed hybrid catfish after 15 days of refrigerated storage. As storage time proceeded, the intensity of spoiled odor, fishy odor, rancid odor, and overall off-odor of all tested muscles tended to increase. Changes in TVB-N content (Figure 1e) and all volatile lipid oxidation concentrations (Figure 4a–i) were associated with these changes. Throughout storage, a consistent trend with the lowest strength of earthy odor was discovered (Figure 5d). When compared to the other muscles, the dorsal muscle had the least earthy odor on day 0 (*p* < 0.05), which was due to a lesser geosmin content in the muscle. Although there was a rising trend of both earthy-causing compounds throughout storage (Figure 3), there was a consistent trend of earthy odor intensity. This could be owing to the masking effect of other odors that grew stronger throughout storage.

## 4. Conclusions

In farmed hybrid catfish, the prevalence of geosmin and 2-MIB compounds, as well as the development of lipid oxidation products, was a major cause of off-odor. Although there was a steady trend of earthy odor intensity during storage, the concentrations of geosmin and 2-MIB in all muscles rose; this phenomenon could be attributable to the masking effect of other off-odors, especially with longer storage. Proteolysis, lipolysis, microbiological degradation, and heme protein dissociation all occurred over time, according to chemical analyses. In farmed hybrid catfish muscle, the rate of lipid oxidation and the formation of volatile compounds was found to be highest in the ventral muscle. The level of spoiled odor, fishy odor, rancid odor, and overall off-odor intensity of all examined muscles tended to increase as storage time progressed. Higher levels of TVB-N and TMA, as well as all other volatile lipid oxidation products, were connected to these changes. Overall, the occurrence and development of off-odor compounds in farmed hybrid catfish muscle during refrigerated storage were determined by the muscle location.

## Figures and Tables

**Figure 3 foods-10-01841-f003:**
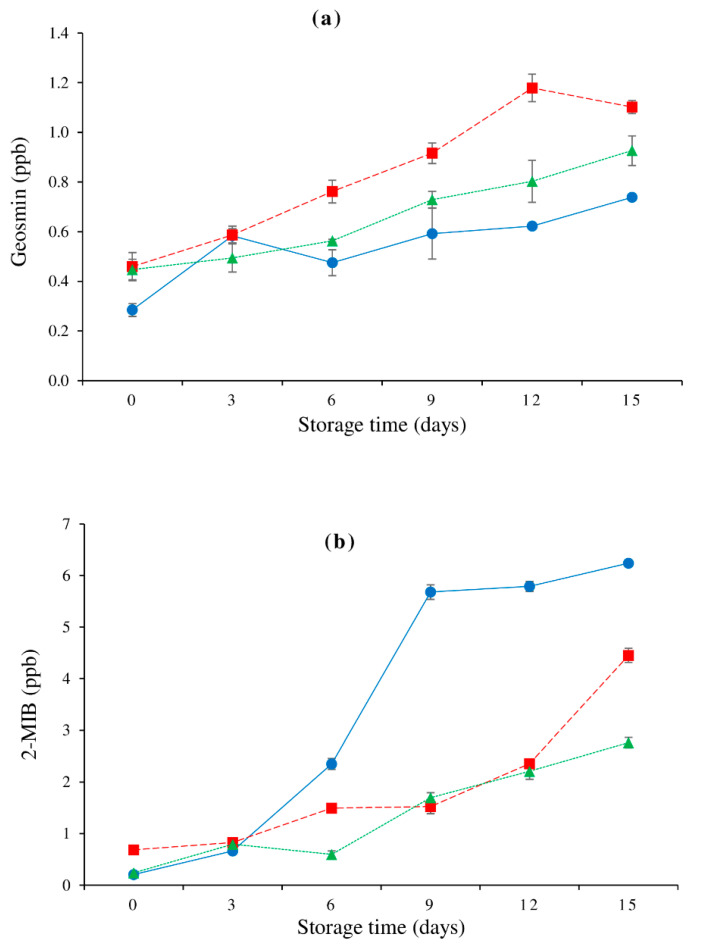
Changes in geosmin (**a**) and 2-MIB (**b**) of farmed hybrid catfish dorsal muscle (―●―), lateral line (---■---), and ventral muscle (⋯▲⋯) during 15 days of refrigerated storage. The bars indicate standard deviation from triplicate determinations.

**Figure 4 foods-10-01841-f004:**
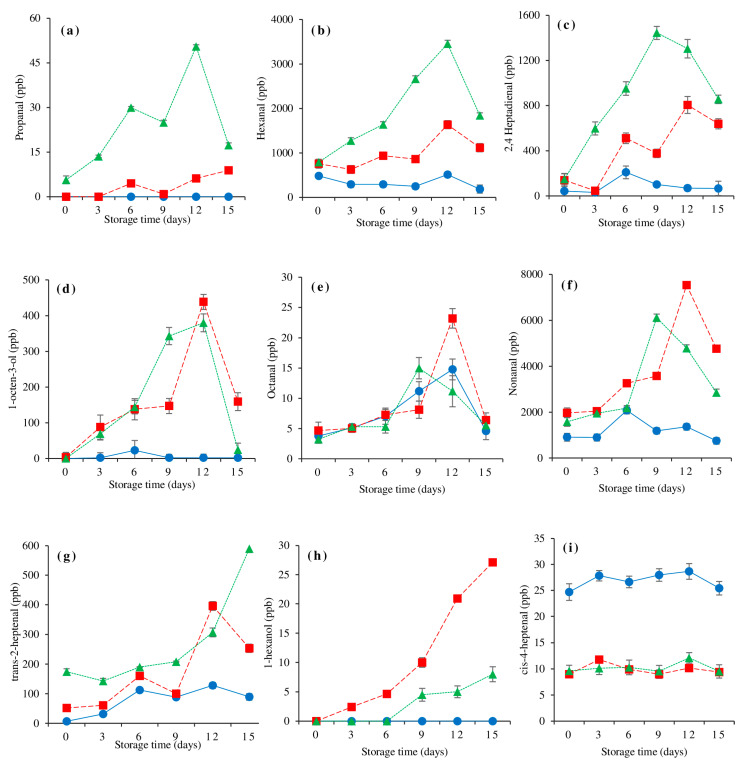
Changes in volatile lipid oxidation products, namely propanal (**a**), hexanal (**b**), 2,4 heptadienal (**c**), 1-octen-3-ol (**d**), octanal (**e**), nonanal (**f**), trans-2-heptenal (**g**), 1-hexanol (**h**) and cis-4-heptenal (**i**) of farmed hybrid catfish dorsal muscle (―●―), lateral line (---■---), and ventral muscle (⋯▲⋯) during 15 days of refrigerated storage. The bars indicate standard deviation from triplicate determinations.

**Figure 5 foods-10-01841-f005:**
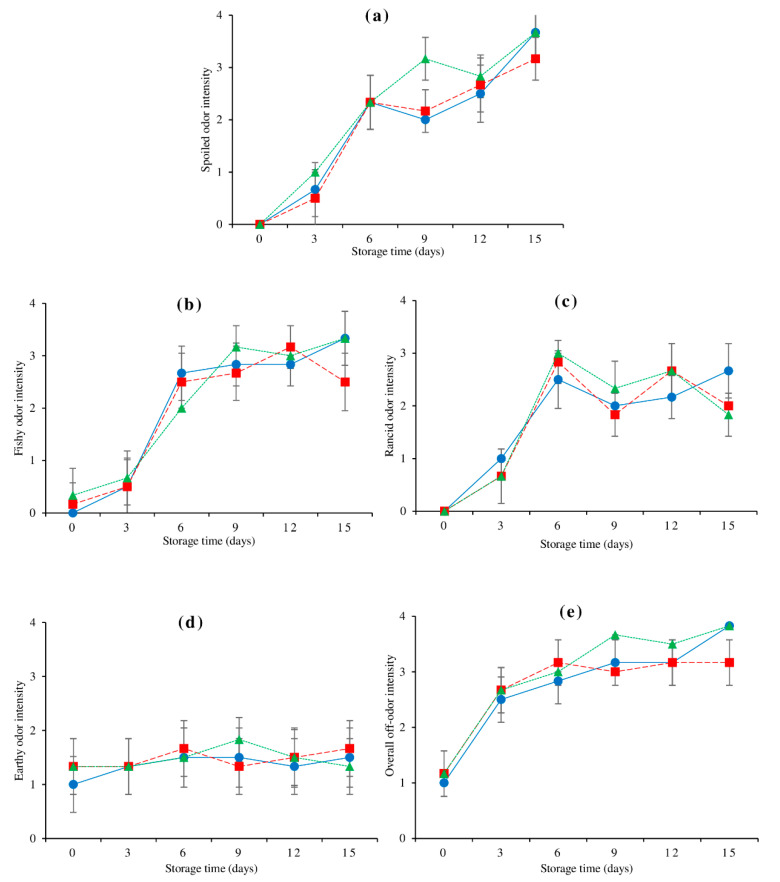
Changes in spoiled odor (**a**), fishy odor (**b**), rancid odor (**c**), earthy odor (**d**), and overall off-odor (**e**) of farmed hybrid catfish dorsal muscle (―●―), lateral line (---■---), and ventral muscle (⋯▲⋯) during 15 days of refrigerated storage. The bars indicate standard deviation from six determinations.

**Table 1 foods-10-01841-t001:** Volatile target compounds in hybrid catfish muscle and their identification parameters.

Volatile Compound	Retention Time (min)	Retention Index		Selected Ion [*m/z*] ^c^	Confirmation ^d^
		Libraries ^a^	Calculation ^b^		
propanal	2.672	798	789	58 *, 29	MS, STD
hexanal	6.237	1083	1085	44 *, 56, 82	MS, STD
cis-4-heptenal	8.774	1240	1247	84 *, 94	MS, STD
octanal	9.495	1289	1295	67 *, 41, 81	MS, STD
trans-2-heptenal	10.047	1323	1334	83 *, 69	MS, STD
1-hexanol	10.396	1355	1359	56 *, 69	MS, STD
nonanal	10.975	1391	1400	58 *, 95, 81	MS, STD
1-octen-3-ol	11.699	1450	1454	72 *, 57	MS, STD
2,4-heptadienal	12.367	1495	1505	81 *	MS, STD
2-MIB	13.694	1592	1612	95 *, 108, 135, 150	MS, STD
geosmin	16.480	1810	1860	112 *, 97, 125	MS, STD

^a^ Retention index libraries: matching with the MS library of NIST (Version 14). ^b^ Retention index calculation: calculated in relation to the retention time of n alkane (C7-C19) series. ^c^ Selected ion: the asterisk, (*) represents quantitative ion. ^d^ Confirmation: STD, comparison of spectra and retention time with commercial standards; MS, tentatively identified by spectra comparison using the MS library of NIST (Version 14).

## Data Availability

Not applicable.

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
