# Peer review of "Occurrence and Development of Off-Odor Compounds in Farmed Hybrid Catfish (Clarias macrocephalus × Clarias gariepinus) Muscle during Refrigerated Storage: Chemical and Volatilomic Analysis"

_foods, 2021, doi:10.3390/foods10081841_

Round 1

Reviewer 1 Report

The Authors report a thorough investigation of chemical and sensory changes during storage of a catfish species. Some remarks intended to improve the quality of the paper:

  • some English corrections are needed (e.g. L15, L136-137, L163-164, L423)
  • abbreviations should be avoided or explained in the abstract
  • figure caption for Figure 1 should be corrected
  • details on the training of the panel mmemers should be delivered
  • L341-342: "This study suggested that the earthy odor problem has always been a key source of negative perception in farmed hybrid catfish species. "- this was not a conclusion of the present study (see Introduction), therefore it should be deleted
  • a major flaw of the research is that its quantitative results are often interpreted as "considerably higher" or "higher" or " tended to rise" and similar statements. In the absence of the analysis of significant differences (e.g. by ANOVA), these comparisons of the results are not scientifically sound.
  • L433-434: "The earthy odor problem caused by geosmin and 2-MIB was found to be the main source of disagreeable perception in fresh hybrid catfish species."- this conclusion is apparently contradicting the previous one in L421-423: "Although there was a rising 421
    trend of both earthy-causing compounds throughout storage (Fig. 3), there was a consistent intensity trend of earthy odor. This could be owing to the masking effect of other odors that grew stronger throughout storage. "

Reviewer 2 Report

The work is interesting but should be better explained. The materials seem to be not in accordance with the traditional conservation of fish. It is not clear what is innovative in this paper.

The introduction is scarce and does not take into consideration the various methods for determining the fish freshness. Just for example: FOOD AND BIOPROCESS TECHNOLOGY (2013) 6: 2190-2195; Food Control (2021) 125,108023; Sensors (2021) 21(4),1373, pp. 1-24; Irish Journal of Agricultural and Food Research (2019) 58(1), pp. 71-80.

While the introduction mentions only 10 publications in the rest of the article, over 60 are cited, I suggest that the number of citations sholuld be drastically reduced eliminating the self-ones.

The aim of the work and its applicative implications should be better explained at the end of the introduction.

Fish were purchased from Thasala market. How do authors know its freshness (hours post mortem)?
Normally fish are stored and transported on ice, while authors report that "Individually packed in plastic bags, the fillets were kept at 4 °C for 15 days". Please justify this choice.

English must necessarily be revised by a native speaker. I would also recommend making phrases impersonal.

Reviewer 3 Report

Occurrence and development of off-odor compounds in farmed hybrid catfish (Clarias macrocephalus × Clarias gariepinus) muscle during refrigerated storage: chemical and volatilomic characterizations

The presented manuscript has interesting results. In my opinion, it would be of interest to the food industry and to the readers of Foods.

There are some comments which are should be considered by Authors:

Title

  1. The title needs some modification the word “characterizations” should not to be written in plural. Also, please consider to find better expression than characterization.

Abstract:

  1. The aim needs reformulation. The expression “look at” is not appropriate. Please revise the aim.
  2. All acronyms must be explained (when used first time). Maybe with exception TBARS.
  3. Line 23, please not use “production”, the word “formation” is much better.
  4. The introduction should be supplemented with some information about nutritional advantages of farmed catfish. When you see on the text you only read about the quality problems of this fish. Please explain why this is fish is worth of consume.

This part of the text should be placed in Line 35.

Materials and Methods

  1. Line 98, please supple the used solvent for lipid extraction from the muscles.
  2. Line 110: Please add on or two sentence for presentation analytical procedure. Also the same for TBARS method Line 119.
  3. 2.8 - I understand that  sensory evaluation was conducted ? Please supply more information. This part methodology is important for understanding the Authors idea. How many times during storages was conducted sensory evaluation ?

Results and Discussion

The discussion of the results generally was well done and was supported by the relevant references.

  1. I have only one suggestion, In the part 3.6. should be mentioned about lipid content in different muscle of farmed hybrid catfish and PUFA content (this is basic information for discussion results of PV and TBARS values).
  2. Please verify the title 3.8.

Conclusions:

In this part the Authors summarized the results. Please add one or two sentences as conclusion. 

Round 2

Reviewer 1 Report

Caption for Figure should still be corrected! (see letters a)-f))

Author Response

Q:Caption for Figure should still be corrected! (see letters a)-f))

A:Caption for Figure 1 was corrected accordingly. Thank you very much.

Reviewer 2 Report

Accept the paper as it is.

Author Response

Thank you very much.

Reviewer 3 Report

The review was properly prepared. The Authors answered on all comments. Thank you.

Author Response

Thank you very much.